# Nirmatrelvir Resistance in an Immunocompromised Patient with Persistent Coronavirus Disease 2019

**DOI:** 10.3390/v16050718

**Published:** 2024-04-30

**Authors:** Chie Yamamoto, Masashi Taniguchi, Keitaro Furukawa, Toru Inaba, Yui Niiyama, Daisuke Ide, Shinsuke Mizutani, Junya Kuroda, Yoko Tanino, Keisuke Nishioka, Yohei Watanabe, Koichi Takayama, Takaaki Nakaya, Yoko Nukui

**Affiliations:** 1Department of Infection Control and Laboratory Medicine, Kyoto Prefectural University of Medicine, Kyoto 602-8566, Japan; k-furu@koto.kpu-m.ac.jp (K.F.); inaba178@koto.kpu-m.ac.jp (T.I.); y-nukui@koto.kpu-m.ac.jp (Y.N.); 2Department of Infectious Disease, Kyoto City Hospital, Kyoto 604-8845, Japan; masa1223@koto.kpu-m.ac.jp; 3Division of Hematology and Oncology, Department of Medicine, Kyoto Prefectural University of Medicine, Kyoto 602-8566, Japan; yniiyama@koto.kpu-m.ac.jp (Y.N.); idedsk@koto.kpu-m.ac.jp (D.I.); mizushin@koto.kpu-m.ac.jp (S.M.); junkuro@koto.kpu-m.ac.jp (J.K.); 4Department of Infectious Diseases, Graduate School of Medical Science, Kyoto Prefectural University of Medicine, Kyoto 602-8566, Japan; y-tanino@koto.kpu-m.ac.jp (Y.T.); din85051@koto.kpu-m.ac.jp (K.N.); nabe@koto.kpu-m.ac.jp (Y.W.); tnakaya@koto.kpu-m.ac.jp (T.N.); 5Department of Pulmonary Medicine, Graduate School of Medical Science, Kyoto Prefectural University of Medicine, Kyoto 602-8566, Japan; takayama@koto.kpu-m.ac.jp

**Keywords:** antiviral drug resistance, SARS-CoV-2, immunocompromised host, nirmatrelvir

## Abstract

Although the coronavirus disease 2019 (COVID-19) pandemic is coming to an end, it still poses a threat to the immunocompromised and others with underlying diseases. Especially in cases of persistent COVID-19, new mutations conferring resistance to severe acute respiratory syndrome coronavirus 2 (SARS-CoV-2) therapies have considerable clinical implications. We present a patient who independently acquired a T21I mutation in the 3CL protease after nirmatrelvir exposure. The T21I mutation in the 3CL protease is one of the most frequent mutations responsible for nirmatrelvir resistance. However, limited reports exist on actual cases of SARS-CoV-2 with T21I and other mutations in the 3CL protease. The patient, a 55 year-old male, had COVID-19 during chemotherapy for multiple myeloma. He was treated with nirmatrelvir early in the course of the disease but relapsed, and SARS-CoV-2 with a T21I mutation in the 3CL protease was detected in nasopharyngeal swab fluid. The patient had temporary respiratory failure but later recovered well. During treatment with remdesivir and dexamethasone, viruses with the T21I mutation in the 3CL protease showed a decreasing trend during disease progression while increasing during improvement. The impact of drug-resistant SARS-CoV-2 on the clinical course, including its severity, remains unknown. Our study is important for examining the clinical impact of nirmatrelvir resistance in COVID-19.

## 1. Introduction

Although the coronavirus disease 2019 (COVID-19) pandemic appears to be ending, it remains a threat to patients with underlying diseases, such as those who are immunocompromised. Persistent COVID-19 in such patients is worrisome, and no established treatment exists for such cases. The persistent presence of severe acute respiratory syndrome coronavirus 2 (SARS-CoV-2) and exposure to therapeutics can lead to the emergence of anti-drug resistance mutations. Currently, available antivirals include RNA-dependent RNA polymerase (RdRp) inhibitors, remdesivir and molnupiravir, and 3-chymotrypsin-like protease (3CL^pro^, also known as the main protease and nonstructural protein 5) inhibitors, including nirmatrelvir (used in combination with ritonavir and marketed as PAXLOVID) and ensitrelvir. The RdRp inhibitors are incorporated into the RNA replicated by RdRp in SARS-CoV-2, inhibiting viral replication by impairing RNA synthesis or causing RNA replication errors. The 3CL^pro^ inhibitors limit viral replication by covalently or non-covalently binding to catalytically important amino acids in the viral M-pro enzyme and functionally inactivating it [1]. Nirmatrelvir is administered orally and has a relative risk reduction of 89% [2]. Nirmatrelvir also reduced the risk of hospitalization or death in patients during the Omicron strain epidemic and after full SARS-CoV-2 vaccination [3,4,5]. In vitro and in vivo studies have demonstrated that mutations are associated with resistance to remdesvir [6,7,8,9,10], and resistance to nirmatrelvir is currently the focus of many studies. Although mutations responsible for nirmatrelvir resistance are becoming apparent [11,12], a few reports have been published on the clinical course of actual cases of SARS-CoV-2 cases with 3CL^pro^ mutations. Therefore, the patient background where 3CL^pro^ mutation is likely to appear and the clinical effects, such as disease severity of nirmatrelvir resistance, remain unclear. Herein, we present a case of SARS-CoV-2 with T21I mutation in 3CL protease after nirmatrelvir exposure.

## 2. Materials and Methods

### 2.1. Cell Culture and Viral Isolation

VeroE6/TMPRSS2 cells (JCRB Cell Bank: JCRB1819) and a Calu-3 human lung cell line (ATCC HTB-55) were cultured and maintained in DMEM supplemented with 10% fetal bovine serum (FBS), 100 U/mL penicillin, 100 μg/mL streptomycin, and 250 ng/mL amphotericin B. Viral isolation was performed as previously described [13]. After filtration using a 0.45 μm membrane filter, each nasopharyngeal swab was adsorbed onto VeroE6/TMPRSS2 cells to isolate the virus for virological use as described below. 

### 2.2. RNA Extraction and Real-Time PCRs

The diagnosis of COVID-19 was confirmed through PCR testing. Briefly, the total RNA was extracted from a 0.25 mL nasopharyngeal swab sample (days 21, 27, 30, and 34) using TRIzol LS reagent and a PureLinkRNA Mini kit (Thermo Fisher Scientific, Waltham, MA, USA), following the manufacturer’s protocols. The extracted RNA was then amplified using real-time RT-PCR with a COBAS SARS-CoV-2 kit on the COBAS 6800 system (Roche, Basel, Switzerland) and with the One Step PrimeScript III RT-qPCR Mix (Takara, Kusatsu, Japan) on a QuantStudio 1 (Thermo Fisher Scientific, Waltham, MA, USA) to quantify the cycle threshold (Ct) value.

### 2.3. Viral Genome Sequencing and Alignment Analysis

For SARS-CoV-2 genotyping, the whole viral genome sequences of the swab samples and isolated viruses (on days 21, 27, 30, and 34) were determined. After library preparation using an NEBNext ARTIC SARS-CoV-2 Companion kit (New England Biolabs, Ipswich, MA, USA), sequencing was performed using a GridION^®^ next-generation sequencer (Oxford NanoporeTechnologies, Oxford, UK) as previously described [13]. To obtain consensus sequences, the sequence data of each sample were aligned to the SARS-CoV-2 Wuhan strain sequence (NC_045512.2) using NanoPipe [14]. Mutations in the whole viral genome were detected with NextClade (https://clades.nextstrain.org/ [accessed 30 April 2024]) using each consensus sequence.

### 2.4. Focus Reduction Neutralizing Test (FRNT)

The nirmatrelvir susceptibility of the two isolated viruses obtained from the swab samples on days 30 and 34 (named YK3 and YK4 here, respectively) was measured using a focus reduction neutralization assay as previously described [13]. Briefly, equal titers of 100 FFU virus were adsorbed onto VeroE6/TMPRSS2 cells in 96 well plates for 1 h at 37 °C. The cells were washed to remove inoculum and overlaid with 1% methylcellulose in DMEM with 0.2% bovine serum albumin containing serial dilutions of nirmatrelvir (starting concentration: 200 μM) dissolved in dimethyl sulfoxide. The cells were cultured for 48 h at 37 °C, washed, and fixed in 4% paraformaldehyde. Immunofluorescence assays were performed with a mouse monoclonal antibody against SARS-CoV-1/2 nucleoprotein clone 1C7C7 (Sigma-Aldrich, St. Louis, MI, USA) and Alexa Fluor 488 secondary antibody. The foci were counted using an inverted fluorescence microscope (Nikon ECLIPSE Ti2 system; Tokyo, Japan). The results were expressed as the 50% focus reduction neutralization titer (FRNT50). These values were calculated using GraphPad Prism 8 software (GraphPad Software, Boston, MA, USA).

## 3. Case Description

We report the case of a 55 year-old man with primary complaints of fever and dyspnea and a history of multiple myeloma since 2018. Eleven courses of chemotherapy had been administered since March 2022. The patient received three doses of the Pfizer BNT162b2 vaccine and one dose of tixiagevimab/cilgavimab.

In December 2022, the patient had a runny nose and fever and underwent a rapid antigen test that was positive for SARS-CoV-2 on day 0. As the patient required no oxygen, he received a 5 day course of nirmatrelvir/ritonavir (NMV/r) (days 4–8) and experienced an improvement in symptomatology. The patient had a fever on day 15, and there was no improvement following antibiotic administration. Therefore, he was admitted on day 21. Upon admission, a reverse transcription–polymerase chain reaction (RT-PCR) test (GeneXpert^®^, Beckman Coulter Inc., Brea, CA, USA) showed that the nasopharyngeal swab was positive for SARS-CoV-2 (cycle threshold (C_t_): 20.5). Genomic sequencing identified the Omicron variant (BF.11.1), and the 3CL^pro^ mutation T21I (C10116T) accounted for 6% of all viruses in nasopharyngeal swabs and 5% of all viruses among isolated viruses. The results of the hematological examination are presented in Appendix A. Computed tomography (CT) performed upon admission revealed multiple nonsegmental ground-glass opacities (GGOs) in the bilateral lung fields (Figure 1A). The clinical course and genomic sequencing results are shown in Figure 1B and Appendix A. The focus reduction neutralization assay showed that the T21I mutation on the BF.11.1 virus increased the 50% inhibitory concentration (IC_50_) values by 3.8 fold (Figure 2A,B).

Owing to his substantial oxygen requirement, the patient was treated with a 7 day course of remdesivir (days 21–27) and a 10 day course of dexamethasone (days 21–30). On day 24, the fever and hypoxemia were resolved, and on day 27, the RT-PCR test was positive for SARS-CoV-2 (C_t_ = 21.7). Genomic sequencing identified viruses with the T21I mutation, accounting for 19% of all viruses in the nasopharyngeal swabs and 80% of all isolated viruses. On day 30, the RT-PCR test showed that the sample was positive for SARS-CoV-2 (C_t_ = 20.7), and genomic sequencing identified viruses with the T21I mutation in 34% of all viruses in the nasopharyngeal swabs and 83% of the isolated viruses, an increase from that observed on day 27.

On day 31, the patient had a fever, and CT demonstrated a new GGO in the upper lung field (Figure 1C), which was treated with another 10 day course of remdesivir (days 32–41) after the relapse of COVID-19. As the patient required no oxygen, dexamethasone was not administered. On day 34, the fever resolved, and an RT-PCR test indicated SARS-CoV-2 (C_t_ = 25.4) infection. Genomic sequencing identified viruses with the T21I mutation in 24% of all viruses in the nasopharyngeal swabs and 2% of the isolated viruses, a decrease from that observed on day 30. His symptoms eventually improved, and he was discharged on day 43. The patient’s clinical course was good after being discharged, and he did not have a COVID-19 relapse even after resumption of chemotherapy for the underlying disease.

All procedures were conducted in accordance with the principles of the Declaration of Helsinki (1964) and its later amendments and approved by the Kyoto Prefectural University of Medicine’s Institutional Review Board (approval number: ERB-C-1435-3). The patient provided written informed consent. Additionally, written consent was obtained from the patient for the publication of this report and the accompanying images.

## 4. Discussion

Immunodeficiency is a risk factor for severe COVID-19, even during the Omicron era [15]. Some patients who are immunocompromised have difficulty acquiring immunity despite vaccination or following infection and illness. Ineffective immune clearance contributes to persistent viral replication in the hosts, providing viruses with many opportunities to mutate. Here, we described the clinical case of nirmatrelvir resistance in a patient who had persistent COVID-19 and was receiving chemotherapy for multiple myeloma. The emergence of viruses resistant to other COVID-19 therapeutics during rituximab and bendamustine therapy or transplantation treatment has been reported previously [9,10,13].

A recent in vitro study demonstrated that the T21I mutation in 3CL^pro^ readily emerges under substantial concentrations of nirmatrelvir. T21 is located around the S1’ sub-pocket of 3CL^pro^ and is involved in hydrogen bonding between 3CL^pro^ and 3CL^pro^ inhibitors. Mutations in T21 result in instability of the hydrogen bonds between them [16]. The T21I mutation alone increased the IC_50_ of nirmatrelvir and ensitrelvir 4.6 and 1.7 fold, respectively [8]. Although the T21I mutation alone only confers low resistance to nirmatrelvir and is presumed to be less of a clinical threat, other mutations derived from T21I or a combination of T21I and other mutations may increase the IC_50_ value of nirmatrelvir and the cross-resistance to other antivirals, such as remdesivir [11,12]. We also isolated BF11.1 viruses harboring T21 or I21 from swabs derived from the patient at different times and compared the IC_50_ values of nirmatrelvir. The T21I mutation on the BF.11.1 virus increased the IC_50_ values 3.8 fold, consistent with the degree of nirmatrelvir resistance when other SARS-CoV-2 lineages acquired the T21I mutation in vitro [11,12]. Based on GISAID data, the probability of having at least one or more mutations in the 3CL protease is 0.5–0.6% during the Omicron strain epidemic, of which T21I is one of the most frequent genetic mutations [17,18]. Additionally, the frequency of this mutation has been reported to show no apparent increasing trend as of 1 year after the launch of nirmatrelvir [17]. However, a few reports exist on actual cases of SARS-CoV-2 with mutations in 3CL^pro,^ including T21I [17], and it remains unclear in which patients the mutations are more likely to appear and how they affect the clinical course, including disease severity. In this case, the patient was treated with NMV/r in the early stages of COVID-19, and the disease quickly resolved. However, the patient experienced a COVID-19 relapse, during which the T21I mutation was detected. A time lag was observed between nirmatrelvir administration and when the T21I mutation became dominant. The clinical condition was stable when the proportion of viruses with the T21I mutation increased. However, the proportion of the wild-type virus increased and was predominant when the clinical symptoms worsened, such as on days 21 and 34 after the initial COVID-19 onset. The reasons for the increase or decrease in the proportion of viruses with the T21I mutation and their relevance to the clinical course are unknown. Thus, the association of 3CL^pro^ mutations, including T21I, with clinical treatment and COVID-19 severity requires further analysis.

The T21I mutation alone does not produce resistance to remdesivir, and its viral replication fitness is comparable with that of the wild type [11]. The P323L mutation in RdRp was also detected during illness (Appendix A). However, this mutation may not have contributed to the remdesivir resistance [19,20]. It has been reported that the mutation of 3CL^pro^ in SARS-CoV-2 reduces the inhibition of interferon production in the host [21]. In this case, the proportion of viruses with the T21I mutation increased with the alleviation of fever or decreased host immune response due to dexamethasone administration and became recessive with dexamethasone discontinuation. Therefore, the T21I mutation may have gradually become recessive because of the immune response. Although the clinical condition was stable when the virus with the T21I mutant increased, remdesivir and dexamethasone administration may have had an effect in this case. However, a reduction in protease activity due to the mutation of 3CL^pro^ in SARS-CoV-2 and a decrease in the severity of illness in affected patients has been reported [22]. Therefore, changes in pathogenicity induced by 3CL^pro^ mutations should be monitored.

This study has some limitations. We did not obtain and evaluate the specimens before nirmatrelvir administration. The T21I mutation emerges spontaneously in patients even before nirmatrelvir administration. Therefore, we could not ascertain that the T21I mutation was induced by nirmatrelvir administration in this case. However, the analysis of the COVIDCG database before the early use authorization of nirmatrelvir showed that the frequency of T21I mutations was extremely rare, being up to 0.24% [23]. Therefore, it is highly possible that the T21I mutation was induced by nirmatrelvir administration.

## 5. Conclusions

We described nirmatrelvir resistance in an immunocompromised patient with persistent COVID-19, highlighting the importance of monitoring 3CL^pro^ mutations during nirmatrelvir therapy. In this case, COVID-19 was persistent. However, the percentage of the virus with the T21I mutation did not correlate with the disease status. Treatment with remdesivir and dexamethasone resulted in improvement and a good clinical course. Further investigation and careful observation of the role of this mutation in the treatment selection, severity of illness, and prognosis of patients with COVID-19 are warranted.

## Figures and Tables

**Figure 1 viruses-16-00718-f001:**
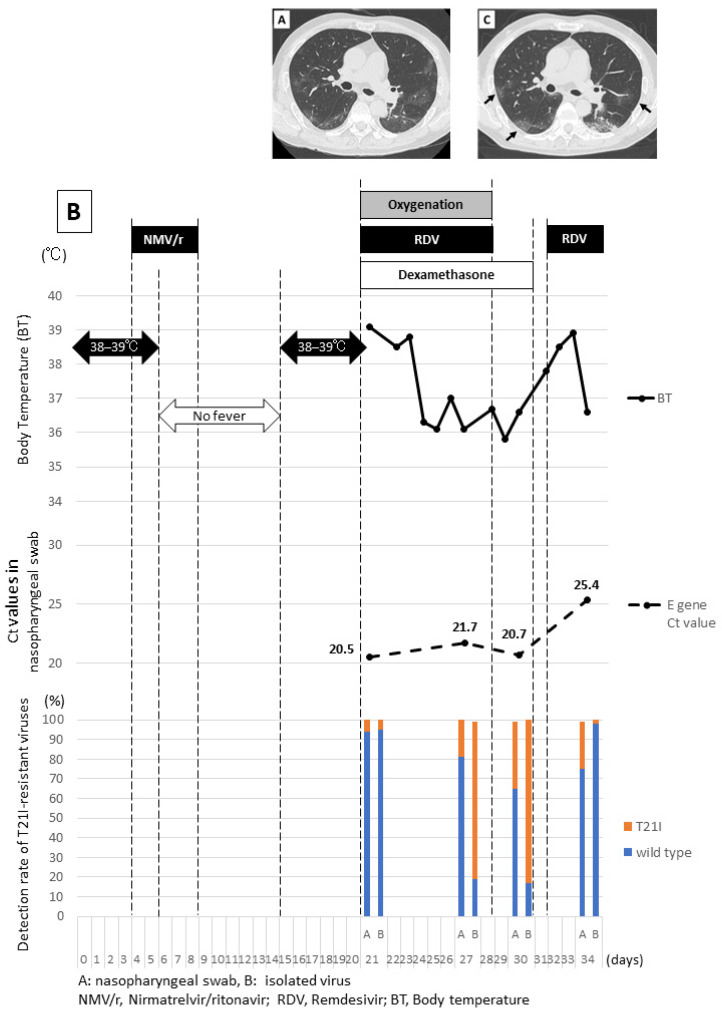
Chest computed tomography, clinical course, and genomic sequencing. (**A**) Axially reconstructed chest CT image 21 days after initial COVID-19 onset. Multiple nonsegmental ground-glass opacities (GGOs) in the bilateral lung fields are shown. (**B**) Timeline of SARS-CoV-2 case and the relationship among NMV/r, remdesivir, and dexamethasone exposure and subsequent development of the 3CL^pro^ T21I mutation. A indicates nasopharyngeal virus; B, isolated virus. The patient was initially administered a 5 day course of NMV/r. Fifteen days after the onset of COVID-19, the patient again had a fever and dyspnea and was admitted on day 21. The patient was treated with a 7 day course of remdesivir and oxygenation and a 10 day course of dexamethasone. On day 24, the patient’s fever and hypoxemia resolved. On day 31, the patient again had a fever and was administered another 10 day course of remdesivir. On day 34, the patient’s fever resolved, and he was discharged on day 43. The Ct values obtained on days 21, 27, 30, and 34 are shown. Genomic sequencing profiles of SARS-CoV-2 were generated using nasopharyngeal swabs and isolated viruses, and the percentages of the wild type and viruses with the 3CL^pro^ mutation T21I (C10116T) are shown. On day 21, viruses with T21I accounted for 6% of all viruses in the nasopharyngeal swabs and 5% of all viruses in the isolated viruses. On days 27 and 30, viruses with T21I accounted for 19% and 34% of all viruses in the nasopharyngeal swabs and 80% and 83% of all viruses in the isolated viruses, respectively. On day 34, viruses with T21I accounted for 24% of all viruses in the nasopharyngeal swabs and 2% of all viruses in the isolated viruses. (**C**) Axially reconstructed chest CT image 33 days after initial COVID-19 onset. The existing GGOs in the lower lobe shadows are partially organized, although multiple new GGOs appear, primarily in the upper lobes of both lungs (black arrows). Abbreviations: 3CL^pro^ = 3-chymotrypsin-like protease; COVID-19 = coronavirus disease 2019; Ct = cycle threshold; CT = computed tomography; NMV/r = nirmatrelvir/ritonavir; SARS-CoV-2 = severe acute respiratory syndrome coronavirus.

**Figure 2 viruses-16-00718-f002:**
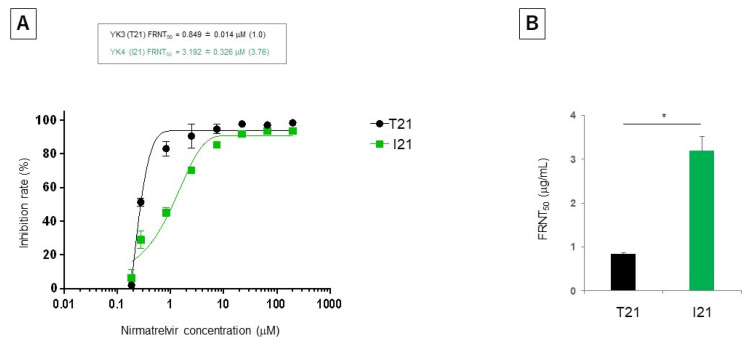
T21I mutation at the 3CL^pro^-induced nirmatrelvir resistance. (**A**) Inhibition rate from the focus numbers at the indicated nirmatrelvir concentration. (**B**) The 50% focus reduction neutralization titer (FRNT50). Data represent the mean ± standard error of the mean (*n* = 3, * *p* < 0.05). The results show that the T21I mutation on the BF.11.1 virus increased the IC_50_ values by 3.8 fold. Abbreviations: 3CL^pro^ = 3-chymotrypsin-like protease; IC_50_ = 50% inhibitory concentration.

## Data Availability

The datasets generated during or analyzed during the current study are available from the corresponding author upon reasonable request.

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
