# Peer review of "Nirmatrelvir Resistance in an Immunocompromised Patient with Persistent Coronavirus Disease 2019"

_viruses, 2024, doi:10.3390/v16050718_

Round 1

Reviewer 1 Report

Comments and Suggestions for Authors

In this manuscript, Yamamoto et al describe a case report of a persistent SARS-CoV-2 infection in an immunocompromised individual. This individual was treated with Paxlovid following COVID-19 diagnosis, and symptoms improved initially, but a fever returned thereafter on day 15. The patient was found to be SARS-CoV-2 positive at this stage, and additional treatment with remdesivir and dexamethasone eventually cleared the virus. Genomic sequencing of samples following the relapse of COVID-19 symptoms revealed the presence of a T21I mutation in the 3CL protease, which had been previously described to confer nirmatrelvir resistance. Neutralization assays comparing a virus with or without the T21I mutation demonstrated that the mutation conferred 3.8-fold resistance to nirmatrelvir.

This case report is well-written and the approach is sound. There has not been a clinical report on T21I to this reviewer’s knowledge, so it is a valuable contribution. It is particularly appreciated that the authors did not overstate their conclusions.

Major comment:

- While it is understood that this is a case report, it is suggested that the addition of a Materials and Methods section with more details on the methodology would be appropriate, rather than having the methods in the figure legends. Notably, details on the sequencing approach, isolation of viruses, and on the neutralization assay would be welcomed. How were the genomic RNA isolated, amplified, and sequenced? How were viruses isolated and would this be expected to have affected their sequences? Which isolated viruses were used for the neutralization assay?

Minor comments:

- Supplementary Table 2 seems to only have the sequencing data for the nasopharyngeal swabs that were sequenced and is missing the isolated viral sequencing data.

·       The raw sequencing data should be deposited to a public repository and should be linked to the manuscript.

·      It would be helpful if the Discussion had some thoughts on what the T21I substitution may be doing since T21 is not a residue that is in direct contact with the nirmatrelvir molecule. There have been a number of recent studies that have explored the function of T21I through viral or biochemical assays.

·       Line 115: Typo for TMPRSS2

·      - Line 204: How can the prevalence be negative?

Author Response

Response to Reviewer 1.

Comments from Reviewer 1.

Comments and Suggestions for Authors

In this manuscript, Yamamoto et al describe a case report of a persistent SARS-CoV-2 infection in an immunocompromised individual. This individual was treated with Paxlovid following COVID-19 diagnosis, and symptoms improved initially, but a fever returned thereafter on day 15. The patient was found to be SARS-CoV-2 positive at this stage, and additional treatment with remdesivir and dexamethasone eventually cleared the virus. Genomic sequencing of samples following the relapse of COVID-19 symptoms revealed the presence of a T21I mutation in the 3CL protease, which had been previously described to confer nirmatrelvir resistance. Neutralization assays comparing a virus with or without the T21I mutation demonstrated that the mutation conferred 3.8-fold resistance to nirmatrelvir.

This case report is well-written and the approach is sound. There has not been a clinical report on T21I to this reviewer’s knowledge, so it is a valuable contribution. It is particularly appreciated that the authors did not overstate their conclusions.

Major comment:

- While it is understood that this is a case report, it is suggested that the addition of a Materials and Methods section with more details on the methodology would be appropriate, rather than having the methods in the figure legends. Notably, details on the sequencing approach, isolation of viruses, and on the neutralization assay would be welcomed. How were the genomic RNA isolated, amplified, and sequenced? How were viruses isolated and would this be expected to have affected their sequences? Which isolated viruses were used for the neutralization assay?

Answer: Thank you for your feedback. We agree with your suggestion and have added a section on Materials and Methods between the Introduction and Case Description in the text. Furthermore, we have deleted part of the legends in Figure 2 (Page 5, Line 157-159). In addition, the order of Reference 13 was changed accordingly, and Reference 14 was added as a new reference (Page 8, Line 321-325). The numbering of references throughout the manuscript has also been changed accordingly.

Minor comments:

- Supplementary Table 2 seems to only have the sequencing data for the nasopharyngeal swabs that were sequenced and is missing the isolated viral sequencing data.

Answer: Thank you for highlighting this important point. We have revised Supplementary Table 2 and added the sequencing data for isolated viruses (Supplementary Table 2).

- The raw sequencing data should be deposited to a public repository and should be linked to the manuscript.

Answer: Thank you for your suggestion. A link to the raw sequencing data has been added in the manuscript (Page 7, Line 263-264).

- It would be helpful if the Discussion had some thoughts on what the T21I substitution may be doing since T21 is not a residue that is in direct contact with the nirmatrelvir molecule. There have been a number of recent studies that have explored the function of T21I through viral or biochemical assays.

Answer: Thank you for the suggestion. A discussion on T21 has been included in the Discussion section and references have been added accordingly (Page 6, Line 203-205 and Page 9, Line 329-331).

-Line 115: Typo for TMPRSS2

Answer: Thank you for pointing out our oversight. After correcting spelling errors, this section has been deleted with the addition of Material and Methods.

- Line 204: How can the prevalence be negative?

Answer: Thank you for pointing out this error. The description has been changed from '-0.24%' to 'up to 0.24%' (Page 7, Line 251).

Reviewer 2 Report

Comments and Suggestions for Authors

Revision manuscript “Nirmatrelvir Resistance in an Immunocompromised Patient with Persistent Coronavirus Disease 2019”

The authors present a well described clinical case of SARS-CoV-2 infection in an immunocompromised patients affected by multiple myeloma. The isolated virus carried a T21I mutation in the 3CL pro conferring resistance to nirmatrelvir. It is not clear the impact of T21I mutation on the clinical course of this immunocompromised patient since he fully recovered after treatment with remdesivir and dexamethasone.

Nirmatrelvir treatment induces rapidly a drug resistance mutation and it seems not effective in eradicating SARS-CoV-2 infection, while remdesivir treatment confirms its value in treating SARS-CoV-2 infection.

Author Response

Response to Reviewer 2.

Comments from Reviewer 2.

Comments and Suggestions for Authors

Revision manuscript “Nirmatrelvir Resistance in an Immunocompromised Patient with Persistent Coronavirus Disease 2019”

The authors present a well described clinical case of SARS-CoV-2 infection in an immunocompromised patients affected by multiple myeloma. The isolated virus carried a T21I mutation in the 3CL pro conferring resistance to nirmatrelvir. It is not clear the impact of T21I mutation on the clinical course of this immunocompromised patient since he fully recovered after treatment with remdesivir and dexamethasone.

Nirmatrelvir treatment induces rapidly a drug resistance mutation and it seems not effective in eradicating SARS-CoV-2 infection, while remdesivir treatment confirms its value in treating SARS-CoV-2 infection.

Answer: Thank you for your feedback. As you suggested, it is unclear to what extent the virus with the T21I mutation affected this patient’s clinical course. It is interesting to note that wild-type was predominant when the patient's symptoms worsened, while the virus with the T21I mutation was predominant when the patient’s condition improved. In fact, on day 49 after the initial onset of the COVID-19, the patient developed fever again, along with slight GGO in the lung field and was treated with remdesivir for 5 days for suspected COVID-19 recurrence (at that time, the use of nirmatrelvir was not permitted in Japan for cases in which a long time had passed since the onset of the disease). However, on day 49 after initial onset of the COVID-19, the Ct value of the PCR from nasopharyngeal wipes did not decrease; therefore, it is unclear whether this was indeed a reoccurrence of COVID-19. At this time, respiratory failure was not complicated, and the fever quickly resolved after restarting remdesivir, and treatment was completed within 5 days. Although genetic analysis was not available at this time, the possibility of a virus with a T21I mutation being involved cannot be ruled out.